# Effect of Alpha-Glucosyl-Hesperidin Consumption on Lens Sclerosis and Presbyopia

**DOI:** 10.3390/cells10020382

**Published:** 2021-02-12

**Authors:** Yosuke Nakazawa, Yuri Doki, Yuki Sugiyama, Ryota Kobayashi, Noriaki Nagai, Naoki Morisita, Shin Endo, Megumi Funakoshi-Tago, Hiroomi Tamura

**Affiliations:** 1Faculty of Pharmacy, Keio University, 1-5-30 Shibako Minato-ku, Tokyo 105-8512, Japan; dokiyuri_0820@keio.jp (Y.D.); yuuki.1998.dragon@keio.jp (Y.S.); ryota_20016koba@keio.jp (R.K.); tago-mg@pha.keio.ac.jp (M.F.-T.); tamura-hr@a7.keio.jp (H.T.); 2Faculty of Pharmacy, Kindai University, 3-4-1 Kowakae, Higashi-Osaka, Osaka 577-8502, Japan; nagai_n@phar.kindai.ac.jp; 3R&D Division, Hayashibara Co., Ltd., Okayama 702-8006, Japan; naoki.morishita@hb.nagase.co.jp (N.M.); shin.endo@hb.nagase.co.jp (S.E.)

**Keywords:** α-glucosyl hesperidin, hesperetin, presbyopia, lens stiffness, TRPV channel, hydrostatic pressure, lens accommodation

## Abstract

Presbyopia is characterized by a decline in the ability to accommodate the lens. The most commonly accepted theory for the onset of presbyopia is an age-related increase in the stiffness of the lens. However, the cause of lens sclerosis remains unclear. With age, water microcirculation in the lens could change because of an increase in intracellular pressure. In the lens, the intracellular pressure is controlled by the Transient Receptor Potential Vanilloid (TRPV) 1 and TRPV4 feedback pathways. In this study, we tried to elucidate that administration of α-glucosyl-hesperidin (G-Hsd), previously reported to prevent nuclear cataract formation, affects lens elasticity and the distribution of TRPV channels and Aquaporin (AQP) channels to meet the requirement of intracellular pressure. As a result, the mouse control lens was significantly toughened compared to both the 1% and 2% G-Hsd mouse lens treatments. The anti-oxidant levels in the lens and plasma decreased with age; however, this decrease could be nullified with either 1% or 2% G-Hsd treatment in a concentration- and exposure time-dependent manner. Moreover, G-Hsd treatment affected the TRPV4 distribution, but not TRPV1, AQP0, and AQP5, in the peripheral area and could maintain intracellular pressure. These findings suggest that G-Hsd has great potential as a compound to prevent presbyopia and/or cataract formation.

## 1. Introduction

The lens is a transparent avascular organ that transmits light from the outside and focuses it on an image in the retina. For distant vision, the lens should become flattered, which is done by the muscular force on the zonules. To allow focusing on near objects, zonular tension is released, and the lens assumes a more rounded shape, which is called the accommodated state. Presbyopia is a disease characterized by loss of near vision as the main symptom and a significant decrease in the quality of life (QOL) and quality of vision (QOV). It is characterized by a sufficient decline in accommodation. With a global increase in life expectancy, the number of people affected by presbyopia is expected to increase. There is no cure for presbyopia, although symptoms can be relieved by the use of presbyopia lenses and contact lenses. Lens sclerosis was reported to be a major inducer of presbyopia, demonstrating that replacement of a presbyopia lens with a soft polymer restores accommodative ability [1], however, the cause of lens sclerosis remains unclear. It has been reported that presbyopia could be an initial symptom of nuclear cataract [2]. The term dysfunctional lens syndrome (DLS), which describes natural lens changes, has become more common in improving patients’ and doctors’ knowledge about lens changes over the years [3]. DLS includes three stages of the lens from presbyopia to cataract. Stage 1 corresponds to presbyopia from 42 to 50 years of age. In stage 2 (50 years or older), contrast and night vision decrease, and the accommodation of the lens is lost. In stage 3 (65 years or older), the lens becomes opaque, and patients present with poor vision quality and full cataract. Cataracts are still the leading cause of blindness worldwide. Therefore, it can be expected that the prevention of presbyopia could lead to a reduction in the number of patients with cataract and blindness.

We previously reported that α-glucosyl-hesperidin (G-Hsd) prevents nuclear cataract formation induced by selenite administration [4]. G-Hsd was synthesized by the glycosylation process with hesperidin (Hsd), which is the major and abundant flavonoid found in citrus fruits. It has a flavanone backbone structure and strong antioxidant activity. Both G-Hsd and Hsd were hydrolyzed by β-glucosidases to hesperetin (Hst), an aglycone of Hsd and G-Hsd. Owing to its 10,000 times greater water solubility, G-Hsd has a higher bioavailability than Hsd and Hst [5]. We demonstrated that oral administration of G-Hsd could prevent lens hardening in 37 weeks old mice which administrated G-Hsd solution for 28 weeks from 9 week old [6]; however, lens sclerosis in 37-week-old mice was not severe.

Lens sclerosis can be initiated by increasing the sodium concentration and intracellular pressure in the lens [7] and controlled by a double-armed feedback pathway that initiates Transient Receptor Potential Vanilloid (TRPV) 1 and TRPV4 [8,9]. We previously reported TRPV1 and TRPV4 subcellular patterns in adult and embryonic mouse lens [10]. The aquaporin (AQP) family is also important for lens water microcirculation. Among the AQP family, AQP0, AQP1, and AQP5 are expressed in lens cells. It was reported that the distribution of TRPV4 in the peripheral fiber cells and AQP5 in the efflux and anterior influx zones altered their subcellular localization in response to mechanical and pharmacologically induced reductions in zonular tension [10,11]. These data suggest that AQPs and TRPVs could exist as a store of channels that could be recruited to the plasma membrane to control the water circulation and intracellular pressure in the lens. In this study, we attempted to elucidate whether administration of G-Hsd could affect the ageing process of the lens and lens elasticity and the effect of G-Hsd treatment on the subcellular distribution of TRPVs and AQPs to meet the requirements of intracellular pressure.

## 2. Materials and Methods

### 2.1. Materials

G-Hsd (including more than 80% α-glucosyl Hsd and less than 20% non-glucosylated Hsd) used in this study were synthesized by Hsd by Hayashibara Co. (Okayama, Japan). C57Black/6JJ (C57BL/6) mice and proper balance chow for mice (CE-2) were purchased from Japan SLC Inc. (Shizuoka, Japan) and Clea Japan Inc. (Tokyo, Japan), respectively. Dithionitrobenzene (DTNB), trichloroacetic acid, and paraformaldehyde (PFA) were obtained from Nacalai Tesque Inc. (Kyoto, Japan). Isoflurane, sodium selenite, reduced glutathione (GSH), ascorbic acid (AsA), 2,6-Dichlorophenolindophenol (DCPIP), and metaphosphoric acid were obtained from Wako Pure Chemical Industries Ltd. (Osaka, Japan). CAT assay kits for catalase (CAT) activity and SOD assay kit-WST for superoxide dismutase (SOD) activity were obtained from Cayman Chemical Inc. (Ann Arbor, MI, USA) and Dojindo Molecular Technologies, Inc. (Kumamoto, Japan), respectively.

### 2.2. Animals

Animals used in this study were housed in a temperature-controlled (23 °C ± 5 °C) environment with a 12-h regular light/dark cycle. Mice were sacrificed using isoflurane (5% inhalational). All animal experiments performed in this study were approved by the Keio University Animal Research Committee (11014-(8)). All animals were treated according to the National Institutes of Health guidelines for the care and use of laboratory animals.

### 2.3. G-Hsd Treatment for Animals

Nine-week-old C57BL/6 mice, either male or female, were separated randomly into six different groups that were administered either water (0% G-Hsd group: *n* = 6), 1% G-Hsd-containing water (1% G-Hsd group: *n* = 8), or 2% G-Hsd-containing water (2% G-Hsd group: *n* = 8) for 16 weeks (short-term exposure: ST) or 49 weeks (long-term exposure: LT) (Figure 1). All mice had unlimited access to chow and drinking water.

### 2.4. Measurement of Lens Elasticity

Lens elasticity was measured using the Softmeasure YAWASA MSES-0512-1 (Tec Gihan Co., Ltd., Kyoto, Japan). The lens was removed immediately after euthanasia and placed on the mount, followed by measurement of the pressure power and dents in the lens. Lens elasticity was calculated using Young’s modulus and the pressure power (mN) required for 15% dents of the lens.

### 2.5. Antioxidant Levels

The levels of GSH and AsA in the lens were measured as previously described with minor modifications [12]. Briefly, to measure the lens concentration of GSH, the lens was homogenized in 0.1 M sodium phosphate buffer (pH 8.0) with 10% trichloroacetic acid and then centrifuged. The absorbance of the supernatant at 412 nm was measured before and 30 min after the addition of DTNB. To measure AsA levels, the lens was homogenized in 0.1 M phosphate-buffered saline (pH 7.4) with metaphosphoric acid. After centrifugation, the supernatant was titrated with DCPIP. Absorbance at 540 nm was measured using an Infinite M200 microplate reader (Tecan Ltd., Mannedorf, Switzerland). CAT activity and SOD activity in the plasma were measured using a CAT assay kit and SOD assay kit-WST, respectively, according to the manufacturer’s protocol.

### 2.6. Immunohistochemistry

The lens was fixed in 0.75% PFA at room temperature for 12 h. Following fixation, all lenses were prepared for cryosectioning using established protocols [10,13]. Cryosections were first treated with 0.1% Triton X-100 for 10 min at room temperature, followed by incubation in the blocking solution (3% bovine serum albumin and 3% normal goat serum in phosphate-buffered saline [PBS]) for 1 h. Sections were incubated in AQP0 antibody (1:100, B-11, Santa Cruz Biotechnology, Dallas, TX, USA), AQP5 antibody (1:100, AB15858, Millipore, Billerica, MA, USA), TRPV1 antibody (1:100, ACC-030, Alomone, Jerusalem, Israel), or TRPV4 antibody (1:100, ab39260, Abcam, Cambridge, UK) antibodies prepared in blocking solution at 4 °C. Slides were then incubated with goat anti-mouse or anti-rabbit Alexa Fluor 488 (1:200, A-11008, Thermo Fisher Science, Waltham, MA, USA) secondary antibody in blocking solution with 4,6-diamidino-2-phenylindole (DAPI, Sigma-Aldrich, St. Louis, MO, USA) to label the nuclei, followed by incubation with wheat germ agglutinin (WGA) Alexa Fluor 594 (1:100, W11262, Thermo Fisher Science) in PBS to label cell membranes. The coverslips were mounted using VectaShield HardSet™ anti-fade mounting medium (Vector Laboratories, Burlingame, CA, USA) and imaged using a laser scanning confocal microscope (Olympus FV1000, Tokyo, Japan).

### 2.7. Statistical Analysis

All data in this study were presented as mean ± SE. Statistical analysis of data was performed using one-way analysis of variance (ANOVA) with a post-hoc Tukey’s multiple comparison test with SPSS software (version 24; IBM Corporation, Armonk, NY, USA). Statistical significance was set at *p* < 0.05.

## 3. Results

### 3.1. G-Hsd Consumption Ameliorated Lens Hardening

Food and water intake in 1% (*v*/*v*) and 2% (*v*/*v*) G-Hsd administration groups at 16 weeks (ST group) and 49 weeks (LT group) was almost identical to that of the control group, and there were no significant differences in body weight among the groups (Figure 2A). Subsequently, the lens stiffness was measured immediately after euthanasia. We measured the lens stiffness using the Yawasa texture and obtained Young’s modulus (Figure 2B). In the ST treatment groups, the mouse lens in the control group was harder than that in the 1% and 2% G-Hsd treatment groups. In the LT treatment group, the mouse control lens was significantly toughened compared to both the 1% and 2% G-Hsd treatment mice lenses. (Figure 2C). Interestingly, the lens stiffness of the 25-week old mice that were administered either 1% or 2% G-Hsd solution was lower than that at 9 weeks of age. These results suggest that G-Hsd treatment could prevent lens sclerosis and improve lens ageing.

### 3.2. G-Hsd Treatment Affected the Antioxidant Ability

Lenses contain high levels of antioxidant compounds, such as GSH and AsA, to prevent oxidative damage. We measured GSH and AsA levels in the lens after G-Hsd treatment. In the control mice (0% G-Hsd treatment), GSH and AsA levels in the lens decreased with age; however, these levels were dose-dependently associated with either 1% or 2% G-Hsd treatment in a concentration-dependent manner (Figure 3A,B). Table 1 lists the lens elasticity, and GSH and AsA levels in the lens with G-Hsd treatment at 16, 28, and 49 weeks in 9-week-old mice (Table 1). Based on the above data, we hypothesized that G-Hsd treatment could inhibit the redux state that decreases with age, we measured plasma antioxidant activities. Without G-Hsd treatment, SOD and CAT activities decreased with age; however, they were also significantly increased by G-Hsd treatment in a dose- and exposure time-dependent manner (Figure 3C,D). These data suggest that oral administration of G-Hsd could prevent age-related changes in redox state levels in the lens and plasma.

### 3.3. G-Hsd Treatment Altered TRPV4 Localization in the Peripheral Fibre Cells of the Lens

Water microcirculation in the lens maintains the lens transparency and accommodation. However, this could change with age because of the increase in sodium concentration and intracellular pressure. To further investigate whether G-Hsd treatment affected the subcellular distribution of AQPs and TRPVs in the peripheral region of the lens, AQPs and TRPVs are known to participate in lens water microcirculation and intracellular pressure. AQP0 and AQP5 in the AQP family and TRPV1 and TRPV4 in the TRPV family are known to be expressed in lens cell fibers [10,14]. To facilitate a comparison between lenses with and without G-Hsd treatment in vivo, peripheral fiber cells of the lens were divided into two zones: the epithelial-fiber differentiation zone and the immature fiber cell zone (Figure 4A). Figure 4 shows the subcellular distribution of AQP0 and AQP5 after 49 weeks of 2% G-Hsd administration. In the peripheral fiber cells of the lenses, we could not detect any changes in the subcellular distribution of AQP0 (Figure 4B,C) and AQP5 (Figure 4D,E), regardless of G-Hsd treatment.

TRPV1 and TRPV4 also play important roles in water circulation and hydrostatic pressure in the lens. The subcellular distribution of TRPV1 and TRPV4 after G-Hsd consumption is shown in Figure 5. TRPV1 was located in the cytoplasm of differentiated and immature cell fibers with or without G-Hsd treatment (Figure 5B,C). TRPV4 in the peripheral areas of the control mouse lens was located in the cytoplasm; however, that in G-Hsd-treated mice was located in the membranous areas (Figure 5D,E). These data suggest that G-Hsd treatment affected the TRPV4 distribution pattern in the peripheral areas of the lens and maintained intracellular pressure.

## 4. Discussion

Presbyopia affects almost everybody by the age of 50 years. As the world’s population is ageing, presbyopia may become one of the most pressing visual concerns of this century, with its global prevalence predicted to increase to 18 billion individuals by 2050 [15,16]. Currently, several surgical techniques are available for the treatment of presbyopia symptoms; however, risks of infection and side effects exist. Therefore, the pharmacological treatment for presbyopia has gained attention recently. The most commonly accepted theory for the onset of presbyopia is an age-related increase in lens stiffness. Tsuneyoshi et al. reported that Pirenoxine eye drops suppressed lens hardening caused by exposure to tobacco smoke in rats and prevented the decrease in objective accommodative amplitude in humans [17]. In this study, we found that oral intake of G-Hsd for 16 or 49 weeks in 9-week-old mice can prevent lens hardening, ameliorate antioxidant states in the lens and plasma, and affect TRPV4 distribution in peripheral fiber cells. Presbyopia and lens hardening are reported to be early signs of nuclear cataracts. It is well known that clouded lenses have low concentrations of GSH and AsA levels compared with transparent lenses. The changes in the redox environment may promote the onset of nuclear cataracts [18]. Our results suggest that the daily intake of G-Hsd is a useful approach for the prevention of both presbyopia and cataract formation.

Based on recent findings, the outflow of water from the central lens nucleus, which occurs across many layers of fiber cells through open gap junction (GJ) channels, generates a substantial hydrostatic pressure in the lens nucleus. We examined Cx50 and Cx46, which are expressed in lens cell fibers and distribution changes with G-Hsd treatment, but did not change their localization (data not shown). In this study, we showed that membrane insertion of lens TRPV4 occurred by G-Hsd treatment for 49 weeks (Figure 5). In other tissues, various external stimuli have been reported to induce membrane insertion of TRPV1 or TRPV4 [19,20,21]. Activation of TRPV1 and TRPV4 channels could involve membrane trafficking in the peripheral regions of the lens in response to hydrostatic pressure control. TRPV1 and TRPV4 are known as osmotic pressure sensors, whose activity is modulated in response to external stimuli such as osmotic stress that controls NKCC1 and Na/K ATPase, respectively [9,22,23]. Recently, Laura et al. reported that Hsd supplementation upregulated the expression of Na+/K+ ATPase in the heart tissue and exerted cardioprotective effects [24]. It has also been reported that oral treatment with Hsd ameliorated lens fluid influx and osmotic imbalance in diabetic cataract lenses [25]. Membrane insertion of TRPV4 and AQP5 was observed in response to accommodation changes [10,11]. As fiber cells differentiate, they lose their cellular organelles and their ability to perform de novo protein synthesis [26,27]. Therefore, differentiating fiber cells or lens epithelial cells tends to synthesize membrane proteins and store them in a cytoplasmic pool in order to meet the physiological needs of the lens. Our results suggest that G-Hsd treatment affected TRPV4 localization to control the Na+/K+ ATPase pump, maintain lens intracellular osmolarity, and prevent lens hardening and presbyopia (Figure 6).

## 5. Conclusions

As we age, we lose our accommodative ability and develop presbyopia, the predominant cause of uncorrected refractive error in the world today. To prevent vision loss and improve the quality of life, there is a strong need to develop novel treatments that can prevent presbyopia. This will benefit mankind immensely. Based on this research, we suggest that G-Hsd has great potential for preventing presbyopia and/or cataract formation after appropriate human clinical trials have been conducted.

## Figures and Tables

**Figure 1 cells-10-00382-f001:**
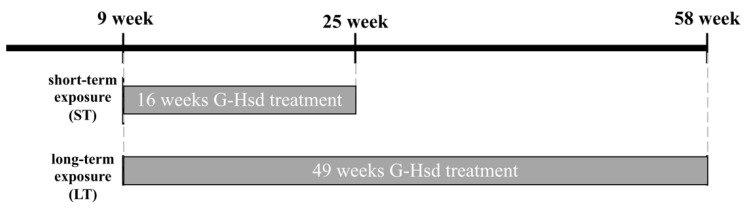
Experimental procedure for G-Hsd treatment. C57BL/6 mice were divided into six groups: treated with 0%, 1%, or 2% G-Hsd solution for 16 weeks (short-term exposure: ST) or 49 weeks (long-term exposure: LT) (*n* = 6 to 8 mice/group).

**Figure 2 cells-10-00382-f002:**
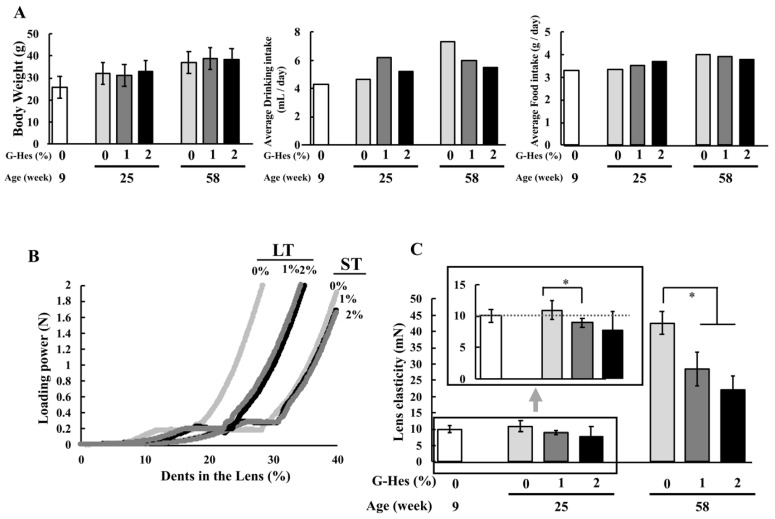
Effect of G-Hsd consumption on lens hardening. Experimental procedure for G-Hsd treatment. C57BL/6 mice were divided into six groups: treated with 0%, 1%, or 2% G-Hsd for 16 weeks (short-term exposure) or 49 weeks (long-term exposure). (**A**) After G-Hsd treatment for 16 or 49 weeks in 9-week old mice, body weight at 25 weeks and 58 weeks of age, water intake and food consumption were measured. (**B**) Young modulus was measured using Softmeasure YAWASA texture. (**C**) Lens elasticity was calculated as the loading power (mN) to achieve 15% dents in the lens. The experiments used three to five independent samples per group. Data are presented as mean ± SEM. * indicates a significant difference versus the control group (*p* < 0.05) (*n* = 6 to 8 mice/group).

**Figure 3 cells-10-00382-f003:**
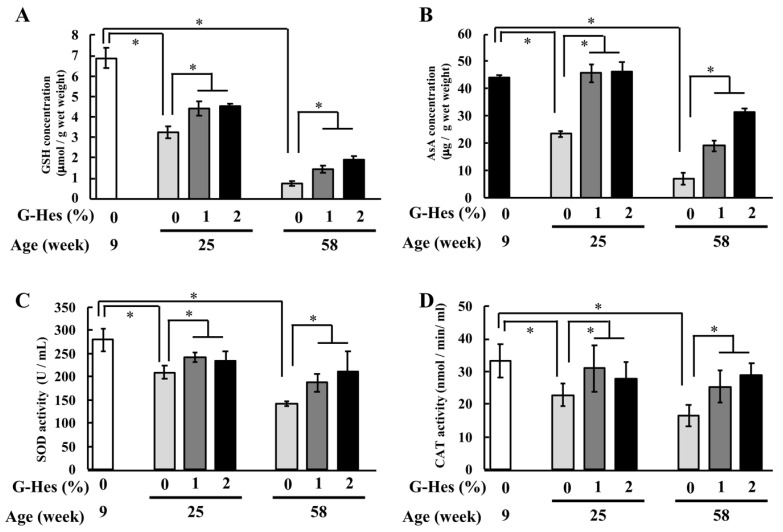
Effect of G-Hsd consumption on the lens and plasma antioxidant levels. Antioxidant levels in the lens and plasma were measured for short-term exposure (ST) and long-term exposure (LT) groups with water, 1% G-Hsd, or 2% G-Hsd. (**A**) Glutathione (GSH) levels and (**B**) AsA levels in the lens were measured (*n* = 4–5). (**C**) SOD activity in the plasma was measured using a Superoxide Dismtase (SOD) assay kit, and (**D**) Catalase (CAT) activity in the plasma was measured using a CAT assay kit (*n* = 6 mice per group). The experiments used three to five independent samples per group. Data are presented as the mean ± SEM. * indicates a significant difference compared to the control group (*p* < 0.05).

**Figure 4 cells-10-00382-f004:**
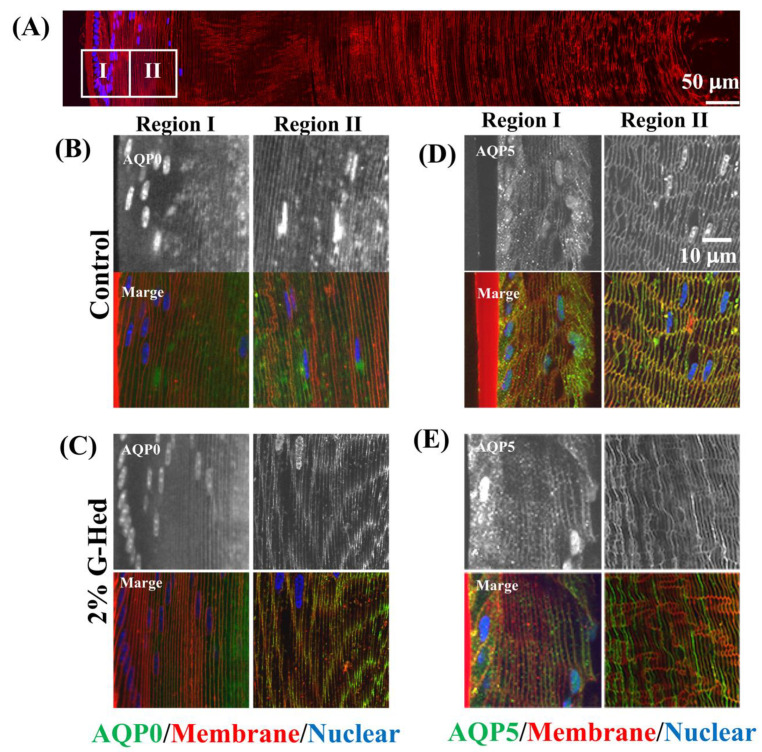
Effect for subcellular distribution of Aquaporins in the lens. After LT administration with 2% G-Hsd solution, subcellular localization of AQP0 and AQP5 in the peripheral regions of the lens was imaged using confocal microscopy. (**A**) A montage of images collected from the lens equator to the core. High power images from the outer 2 regions (boxes) were obtained. (**B**,**C**) High power images of sections labeled with AQP0 from (**B**) control mice lens and (**C**) 2% G-Hsd for 49-week exposed mice lens. Upper panels were images of AQP0 signal alone, and lower panels were merged images with AQP0, WGA (membrane), and DAPI (nuclear). (**D**,**E**) High power images of sections labeled with AQP5 from (**D**) control mice lens and (**E**) 2% G-Hsd for 49-week exposed mice lens. Upper panels were images of AQP5 signal alone, and lower panels were merged images with AQP5, WGA 5, and DAPI. The experiments used three to five independent samples per group.

**Figure 5 cells-10-00382-f005:**
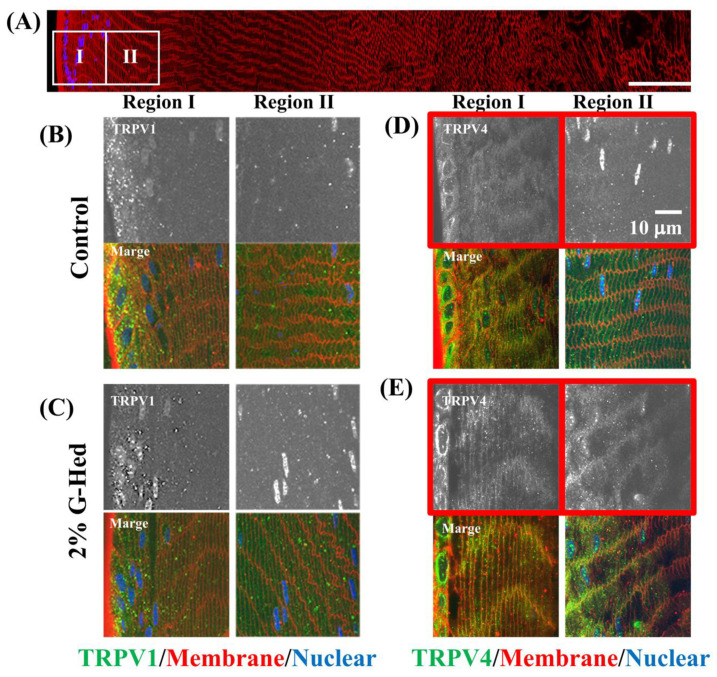
Effect for distribution of TRPV channels in the peripheral region of the lens. After LT treatment of G-Hsd, subcellular localization of TRPV1 and TRPV4 in the peripheral regions of the lens was measured using confocal microscopy. (**A**) A montage of images collected from the lens equator to the core. High power images from the outer 2 regions (boxes) were obtained. (**B**,**C**) High magnified images of sections labeled with TRPV1 from (**B**) control mice lens and (**C**) 2% G-Hsd for 49-week exposed mice lens. Upper panels are images of TRPV1 signal alone, and lower panels were merged images with TRPV1, WGA, and DAPI. (**D**,**E**) High magnified images of sections labelled with TRVP4 from (**D**) control mice lens and (**E**) 2% G-Hsd for 49-week exposed mice lens. Upper panels were images of TRPV4 signal alone, and lower panels were merged images of TRPV4, WGA, and DAPI. The experiments used three to five independent samples per group.

**Figure 6 cells-10-00382-f006:**
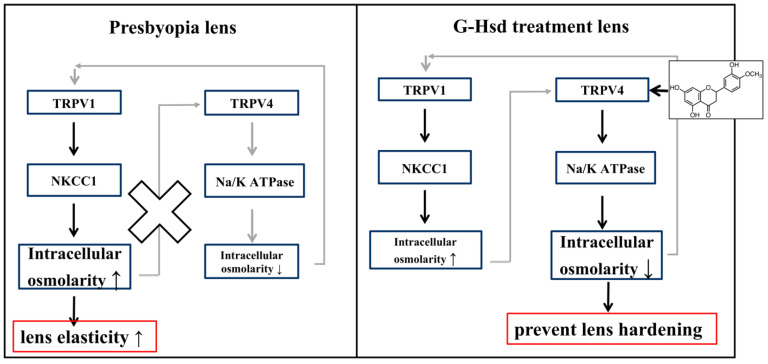
Inhibitory mechanisms of presbyopia onset by G-Hsd consumption. G-Hsd consumption affects TRPV4 localization and activates TRPV4 and Na/K ATPase pump, which decrease the intracellular osmolarity of the lens and prevent lens hardening.

**Table 1 cells-10-00382-t001:** The lens elasticity and GSH and Ascorbic Acid (AsA) levels in the lens for 16 weeks, 28 weeks, and 49 weeks consumption of G-Hsd.

Age	9 weeks	25 weeks (9 + 16)			37 weeks (9 + 28) (#)			58 weeks (9 + 49)		
G-hsd	0% (control)	0%	1%	2%	0%	1%	2%	0%	1%	2%
Lens elasticity (mN)	10.0 ± 1.08	11.0 ± 1.57	8.93 ± 0.66	7.75 ± 2.96	11.6 ± 0.39	7.30 ± 0.78	7.95 ± 0.44	42.6 ± 3.44	28.6 ± 5.14	22.1 ± 4.23
GSH (mg/g)	6.87 ± 0.50	3.50 ± 0.29	4.41 ± 0.33	4.53 ± 0.10	1.06 ± 0.11	3.24 ± 0.58	4.12 ± 0.42	0.73 ± 0.11	1.44 ± 0.17	1.91 ± 0.16
AsA (mg/g)	44.2 ± 0.80	23.4 ± 1.26	45.8 ± 3.26	46.1 ± 3.47	17.1 ± 1.20	24.4 ± 1.30	26.2 ± 4.00	7.01 ± 2.06	19.0 ± 2.00	31.4 ± 1.13

^(#)^ Data from the accepted paper from Nakazawa et al. [6].

## Data Availability

The data presented in this study are available on request from the corresponding author.

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
