# Peer review of "Effect of Alpha-Glucosyl-Hesperidin Consumption on Lens Sclerosis and Presbyopia"

_cells, 2021, doi:10.3390/cells10020382_

Round 1
Reviewer 1 Report
The paper by Nakazawa and co-workers describes the potential use of G-Hsd as antipresbyopia agent and, eventually, as anti-cataract agent. Despite the matter is of certain interests to the readers and some of the results shown could be important for some specific advancement in the field, the paper is penalized by several major flaws needing a deep and accurate revision before acceptance.
1) The pathologic relationship between prebyopia and cataract is assumed by the authors on the basis of a quite old and outdated bibliographic reference. Please introduce the most recent references corroborating the hypotesis.
2) It is unclear to me which was the title of the reference compound used during in vivo experiments. It seems it could be 80% form the section 2.1. In this case authors should provide clear evidences about the composition of the material concerning the unknown 20%. Otherwise is really hard to imagine the pharmacological effects measured are attributable to the G-Hsd.
3) The authors state that nine weeks mice were separated randomly in six different groups (protocols in the main text, please correct). The questions are: how many mice in each group? Did the authors consider sex differences? Is, for instance, the lens stiffeness different form male to female mice? Could you provide references about?
4) G-Hsd was adminidtered via a watwer solution but animals had unlimited access to chow and drinking water. Which is the daily dose given? This makes a great difference in terms of potential translability of the research.
5) The authors performed experiments about lens elasticity, antioxidant levels in lens and lens immunoistochemisty. How could they perform three different experiments every single animal since they declare that statistics were performed on a n=6-8 mice/group?
6) The authors found G-Hsd altered TRPV4 localization in the epripheral fibre cells corralating this issue with the intraocular pressure control. Did they check intraocular pressure variations? It should be the very first exepriment to be performed to verify the hypotesis
7) Language editing is required throughout the paper.
Author Response
Answers to Reviewer #1
Thank you very much for reviewing our manuscript. As indicated in the responses that follow, we have taken all of these comments into consideration in the revised version.
The paper by Nakazawa and co-workers describes the potential use of G-Hsd as antipresbyopia agent and, eventually, as anti-cataract agent. Despite the matter is of certain interests to the readers and some of the results shown could be important for some specific advancement in the field, the paper is penalized by several major flaws needing a deep and accurate revision before acceptance.
Response: Thank you for reviewing our manuscript. We are grateful for your valuable comments, which have helped us improve the quality of our manuscript considerably.
1) The pathologic relationship between prebyopia and cataract is assumed by the authors on the basis of a quite old and outdated bibliographic reference. Please introduce the most recent references corroborating the hypotesis.
Response: Thank you for your suggestion. We have added the following section about dysfunctional lens syndrome to the Introduction section:
Dysfunctional lens syndrome (DLS), which describes natural lens changes, has become more common in improving patients’ and doctors’ knowledge about lens changes over the years [3]. DLS includes three stages of the lens from presbyopia to cataract. Stage 1 corresponds to presbyopia from 42 to 50 years of age. In stage 2 (50 years or older), contrast and night vision decreases, and accommodation of the lens is lost. In stage 3 (65 years or older), the lens becomes opaque, and patients have poor vision quality and full cataract. (Line 44 – 50)
[3] J. Fernández, M. Rodríguez-Vallejo, J. Martínez, A. Tauste, D.P. Piñero, From Presbyopia to Cataracts: A Critical Review on Dysfunctional Lens Syndrome, J Ophthalmol 2018 (2018) 4318405. 10.1155/2018/4318405.
2) It is unclear to me which was the title of the reference compound used during in vivo experiments. It seems it could be 80% form the section 2.1. In this case authors should provide clear evidences about the composition of the material concerning the unknown 20%s. Otherwise is really hard to imagine the pharmacological effects measured are attributable to the G-Hsd.
Response: Thank you for your comment. In this study, we used a G-Hsd compound that was enzymatically synthesised from hesperidin purified from orange (Citrus aurantium) and dextrin. It contains monoglucosyl hesperidin (more than 80%) and nonglucosylated hesperidin (less than 20%), which was confirmed by high-performance liquid chromatography. G-Hsd is hydrolysed by a-glucosidase in the intestine from G-Hsd to Hsd. All Hsd-related compounds in the blood were Hsd and its aglycone, Hst. Alpha-glucosyl hesperidin has been approved by the Department of Consumer Affairs Agency, Government of Japan.
We have added the following text in the Materials and Methods section:
“G-Hsd (include more than 80% a-glucosyl Hsd and less than 20% nonglucosylated Hsd) used in this study was synthesised by Hsd by Hayashibara Co.” (Line 76 – 77)
3) The authors state that nine weeks mice were separated randomly in six different groups (protocols in the main text, please correct).
Response: Thank you for your comment. We have changed it accordingly.
The questions are: how many mice in each group? Did the authors consider sex differences? Is, for instance, the lens stiffeness different form male to female mice? Could you provide references about?
Response: Thank you for your comment. In the present study, we used 44 male and female mice. A meta-analysis reported that there were no significant sex differences in accommodative ability.
There were six mice in the control group and eight mice in the 1% and 2% G-Hsd treatment groups for ST exposure and LT exposure. Four lenses from different animals were used for immunohistochemistry, and the remaining lenses were used to measure GSH and AsA levels after lens elasticity measurement.
We made the following changes in the Materials and Methods section:
‘Nine-week-old C57BL/6 mice, either male or female, were separated randomly into six different groups that were given either water (0% G-Hsd group: n = 6), 1% G-Hsd-containing water (1% G-Hsd group: n = 8), or 2% G-Hsd-containing water (2% G-Hsd group: n = 8) for 16 weeks (short-term exposure: ST) or 49 weeks (long-term exposure: LT) (Scheme 1). All mice had unlimited access to chow and drinking water.’ (Line 95 – 99)
4) G-Hsd was adminidtered via a watwer solution but animals had unlimited access to chow and drinking water. Which is the daily dose given? This makes a great difference in terms of potential translability of the research.
Response: Thank you for your valuable comment. We did not measure the exact volume of water drunk by each animal. In this report, the mice drank about 4 to 5 ml a day on average. We ensured that they took about 1.5 g/kg (1% G-Hsd group) and 3.0 g/kg (2% G-Hsd group) of G-Hed compounds. In addition, we tried to obtain a G-Hsd solution for mice via feeding tubes orally to determine the exact amount they consumed. Thank you once again for your suggestion.
5) The authors performed experiments about lens elasticity, antioxidant levels in lens and lens immunoistochemisty. How could they perform three different experiments every single animal since they declare that statistics were performed on a n=6-8 mice/group?
Response: Thank you for your comment. We performed more than three independent experiments in this report by more than two scientists and/or students. We have put in the figure legends: ‘The experiments used three to five independent samples per group. (Figure 2, line 161; Figure 3, line 184; Figure 4, line 226; Figure 5, line 237)
6) The authors found G-Hsd altered TRPV4 localization in the epripheral fibre cells corralating this issue with the intraocular pressure control. Did they check intraocular pressure variations? It should be the very first exepriment to be performed to verify the hypotesis
Response: Thank you for your comment. The intraocular pressure was not measured after G-Hsd treatment. In this report, we considered changing the intracellular pressure in the lens, but not intraocular pressure after G-Hed consumption. However, there are some reports that G-Hsd and hesperidin have anti-glaucoma potential. We will try to measure intraocular pressure after G-Hsd consumption.
7) Language editing is required throughout the paper.
Response: Thank you very much for carefully reviewing our manuscript. We have had our manuscript rechecked by Editage, an English language editing service.
Reviewer 2 Report
The manuscript by Nakazawa et. al. titled: "Effect of Alpha-glucosyl-hesperidin consumption on lens sclerosis and presbyopia" details a study directed at investigating the use of a-glucosyl-hesperidin (G-Hsd) in treating the condition presbyopia. The authors discover that G-Hsd-treated lenses at 1% or 2% on nine-week old mice for 16 wks (ST) or 49 wks (LT) are significantly less hardened than their controls. The functional probing of this difference was through identifying distribution differences in ion channels (AQs and TRPV channels) in treated vs control lenses. They found that TRPV4 is notably distributed differently in the treated lenses.
Author Response
Answers to Reviewer #2
The manuscript by Nakazawa et. al. titled: "Effect of Alpha-glucosyl-hesperidin consumption on lens sclerosis and presbyopia" details a study directed at investigating the use of a-glucosyl-hesperidin (G-Hsd) in treating the condition presbyopia. The authors discover that G-Hsd-treated lenses at 1% or 2% on nine-week old mice for 16 wks (ST) or 49 wks (LT) are significantly less hardened than their controls. The functional probing of this difference was through identifying distribution differences in ion channels (AQs and TRPV channels) in treated vs control lenses. They found that TRPV4 is notably distributed differently in the treated lenses.
Response: Thank you very much for reviewing our paper.
Round 2
Reviewer 1 Report
First of all, i would like to thanks authors for the efforts made in the revision process and, in particular, for the politeness and correctness of their replies. This is a very rare attitude. The authors have properly replied to my comments and amended accordingly the main text. Collectively, the quality of the paper has been increased and now it is acceptable for publication in Cells.